



**Technical note: on LA–ICP-MS U–Pb dating of unetched and etched apatites**
Fanis Abdullin et al.: LA–ICP-MS U–Pb dating of apatites
Fanis Abdullin[1], Luigi Solari[2], Jesús Solé[3], Carlos Ortega-Obregón[2]
[1]CONACyT–Centro de Geociencias, Campus Juriquilla, UNAM, Querétaro, 76230, Mexico
[2]Centro de Geociencias, Campus Juriquilla, UNAM, Querétaro, 76230, Mexico
[3]LANGEM, Instituto de Geología, UNAM, Ciudad Universitaria, CDMX, 04510, Mexico
**Correspondence:** Fanis Abdullin (fanis@geociencias.unam.mx)
**Abstract**
The same unetched and chemically etched apatites from five rock samples were dated with U–Pb
using laser ablation inductively coupled plasma mass spectrometry. The objective of this study is
to demonstrate whether or not the etching, needed for the apatite fission track analysis, impact on
the obtaining of apatite U–Pb ages. The results of this experiment indicate that the etching has no
effect on the determination of apatite U–Pb ages by the laser ablation inductively coupled plasma
mass spectrometry technique. Thus, laser ablation inductively coupled plasma mass spectrometry
may be used safely for simultaneous apatite fission track *in-situ* and U–Pb double dating.
**Short summary**
Unetched and etched apatites of five samples were dated by U–Pb with laser ablation inductively
coupled plasma mass spectrometry. Our experiment demonstrates that the etching, needed for the
apatite fission track dating, has no important effect on the obtaining of U–Pb ages; and therefore,
the laser ablation-based technique can be used for apatite fission track and U–Pb double dating.



## 1    Introduction

Apatite, $Ca_5(PO_4)_3[F,Cl,OH]$, is the most common phosphate mineral in the Earth's crust and can be found in practically all igneous and metamorphic rocks, as well as in many ancient and recent sediments and in certain mineral deposits (Piccoli and Candela, 2002; Morton and Yaxley, 2007; Webster and Piccoli, 2015). This accessory mineral is often used as a natural thermochronometer (i.e., for fission track, helium and U–Pb dating; e.g., Zeitler et al., 1987; Wolf et al., 1996; Ehlers and Farley, 2003; Hasebe et al., 2004; Donelick et al., 2005; Chew and Donelick, 2012; Chew et al., 2014; Cochrane et al., 2014; Liu et al., 2014; Spikings et al., 2015; Glorie et al., 2017).

Presently, apatite fission track (AFT) ages may be obtained rapidly by using LA–ICP-MS (laser ablation inductively coupled plasma mass spectrometry) for direct measurement of "parent nuclides", i.e., $^{238}U$ levels (Cox et al., 2000; Svojtka and Košler, 2002; Hasebe et al., 2004, 2009; Donelick et al., 2005; Vermeesch, 2017). In addition, the LA–ICP-MS-based technique allows to date apatites simultaneously by AFT and U–Pb (e.g., Chew and Donelick, 2012; Liu et al., 2014; Glorie et al., 2017; Bonilla et al., 2020; Nieto-Samaniego et al., 2020). After chemical etching of apatites, a smaller volume of ablated material is analyzed with LA–ICP-MS. Therefore, there is a doubt on the application of such double dating technique. The question is how chemical etching, required for the AFT dating, may influence on the obtaining of apatite U–Pb ages? To respond to this question, the same unetched and etched apatite crystals from five experimental samples were dated by LA–ICP-MS U–Pb. The chosen rock samples have either emplacement or metamorphic ages varying from the Early Cretaceous to the Neoproterozoic (for details, please see Table 1).

--- **Table 1** ---



**2      Brief description of samples**

2.1      OV-0421 (Tres Sabanas Pluton, Guatemala)

This sample is a two mica-bearing deformed granite belonging to the Tres Sabanas Pluton, which
is located NW of Guatemala City, Guatemala. For sample OV-0421, an emplacement age of 115
± 4 Ma was proposed based on zircon U–Pb data (Torres de León, 2016). A cooling age of 102 ±
1 Ma, obtained with K–Ar (on biotite concentrate), has also been reported by the same author.

2.2.      MCH-38 (Chiapas Massif Complex, Mexico)

MCH-38 is an orthogneiss from the Permian Chiapas Massif Complex. This rock was sampled to
the west of Unión Agrarista, the State of Chiapas, southeastern Mexico. There is no reported age
for this sample. Some zircon U–Pb dates obtained for the Chiapas Massif Complex (Weber et al.,
2007, 2008; Ortega-Obregón et al., 2019) suggest that a Lopingian (260–252 Ma) crystallization
or metamorphic age may be assumed for sample MCH-38.

2.3      TO-AM (Totoltepec Pluton, Mexico)

TO-AM is a granitic rock, sampled ca. 5 km west of Totoltepec de Guerrero, the State of Puebla,
southern Mexico. There is no reported radiometric data for sample TO-AM. Previous geological
studies indicate that the Pennsylvanian–Cisuralian Totoltepec Pluton was emplaced over a ca. 20
million year period (from 306 ± 2 to 287 ± 2 Ma; e.g., Kirsch et al., 2013).

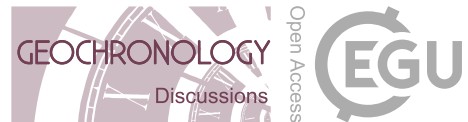


2.4    CH-0403 (Altos Cuchumatanes, Guatemala)

CH-0403 was collected 5 km ESE of Barillas, in the Altos Cuchumatanes, Guatemala. It consists
of a gray to green granodiorite. Five zircon aliquots of sample CH-0403 were dated using isotope
dilution thermal ionization mass spectrometry, yielding a lower intercept date of $391 \pm 8$ Ma that
is interpreted as its approximate crystallization age (Solari et al., 2009).

2.5    OC-1008 (Oaxacan Complex, Mexico)

This sample is a paragneiss from the Grenvillian Oaxacan Complex, southern Mexico. OC-1008
was collected in the federal road which connects Nochixtlán to Oaxaca. It was demonstrated that
this sample underwent "dry" granulite facies metamorphism at $990 \pm 10$ Ma (Solari et al., 2014).


**3    Analytical procedures**

Apatites were concentrated using conventional mineral separation techniques. Nearly 300 apatite
grains, extracted from each rock sample, were mounted with EpoFix™ in a 2.5 cm diameter ring.
Apatites were mounted with their surfaces parallel to the crystallographic $c$-axis. Mounted grains
were polished to expose their internal surfaces (i.e., up to $4\pi$ geometry). For our experiment, only
"sterile" and complete crystals, without visible inclusions and other defects such as cracks, were



gently selected. Sample preparation was performed at Taller de Molienda, Centro de Geociencias
(CGEO), Campus Juriquilla, Universidad Nacional Autónoma de Mexico (UNAM).

94         Single spot analyses were performed with a Resonetics RESOlution™ LPX Pro (193 nm,

ArF excimer) laser ablation system, coupled to a Thermo Scientific iCAP™ Qc quadrupole ICP-
MS at Laboratorio de Estudios Isotópicos (LEI), CGEO, UNAM. During this experimental work,
LA–ICP-MS-based sampling was performed exactly in central parts of the selected apatite grains
before and after chemical etching (in 5.5M $HNO_3$ at 21 °C for 20 s to reveal spontaneous fission
tracks), as shown schematically in Fig. 1. The LA–ICP-MS protocol used for apatite analyses, as
given in Table 2, was established on the basis of numerous experiments carried out at LEI during
the past five years, and can be used for U–Pb and fission track double dating plus multielemental
analysis (Abdullin et al., 2018; Ortega-Obregón et al., 2019). Corrected isotopic ratios and errors
were calculated using Iolite (Paton et al., 2011) and the VizualAge data reduction scheme (Petrus
and Kamber, 2012). UcomPbine (Chew et al., 2014) was used to model $^{207}Pb/^{206}Pb$ initial values
and thus force a $^{207}Pb$ correction that considers the common Pb (non-radiogenic Pb) incorporated
by apatite standards at the moment of their crystallization (see also Ortega-Obregón et al., 2019).
The "First Mine Discovery" apatite from Madagascar, with a mean $^{206}U–^{238}Pb$ age of ca. 480 Ma
(Thomson et al., 2012; Chew et al., 2014), was used as a main reference material. The results for
measured isotopes using the NIST-612 glass (Pearce et al., 1997) were normalized using $^{43}Ca$ as
an internal standard and taking an average CaO content of 55% (i.e., for F-apatites).

111                              --- **Figure 1** ---

112                              --- **Table 2** ---





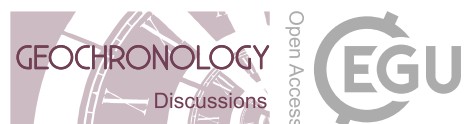

**4     Results**

Tera and Wasserburg Concordia diagrams (T–W; Tera and Wasserburg, 1972) are used in apatite
U–Pb dating, because the LA–ICP-MS-derived U–Pb results are generally discordant. The lower
intercept in the T–W plot is considered as a "mean" apatite U–Pb age that should have geological
significance (crystallization or cooling age, or the ages of mineralization or metamorphic event).
Apatite U–Pb ages were calculated with IsoplotR (Vermeesch, 2017, 2018) and described below.
Detailed information on our U–Pb experiments is given in Table S1 in the Supplement.

4.1     OV-0421

For sample OV-0421, 41 unetched apatites analysed yielded a lower intercept age of 106 ± 4 Ma
with a mean square weighted deviation (MSWD) of 1.07, passing the chi-squared probability test
with the $P(\chi^2)$ value of 0.35 (see Fig. 2). Virtually the same U–Pb age, 107 ± 5 Ma, was obtained
after chemical etching of the same apatite crystals, yielding a MSWD of 1.13 and a $P(\chi^2)$ of 0.27.
Both these apatite U–Pb ages lie between the zircon U–Pb age of 115 ± 4 Ma (i.e., crystallization
age) and the biotite K–Ar date of 102 ± 1 Ma (i.e., cooling age), which were previously obtained
for the same granite sample by Torres de León (2016).

4.2.    MCH-38

For orthogneiss sample MCH-38, the lower intercept in T–W yielded a U–Pb age of 245 ± 6 Ma
(obtained from 41 unetched grains) with a MSWD of 0.28 and a $P(\chi^2)$ of 1. Etched apatite grains





from MCH-38 yielded an age of 240 ± 4 Ma with a MSWD of 0.36 and a $P(\chi^2)$ of 1 (Fig. 2). Our
U–Pb ages are in close agreement with geochronological data reported from the Permian Chiapas
Massif Complex in previous studies (Damon et al., 1981; Torres et al., 1999; Schaaf et al., 2002;
Ortega-Obregón et al., 2019). For instance, Torres et al. (1999) compiled biotite K–Ar ages, most
of which lie within the Early–Middle Triassic period. Triassic cooling ages in the Chiapas Massif
Complex were also detected by Rb–Sr in mica–whole rock pairs that range from 244 ± 12 to 214
± 11 Ma (Schaaf et al., 2002).

4.3    TO-AM

Unetched apatites (32 crystals; Fig. 2) from granite TO-AM yielded a lower intercept date of 303
± 5 Ma with a MSWD of 0.6 and a $P(\chi^2)$ of 0.96. After etching, a slightly younger age of 299 ± 3
Ma was obtained, with a MSWD of 0.89 and a $P(\chi^2)$ of 0.65. These apatite U–Pb dates are in line
with the zircon U–Pb ages of 306 ± 2 to 287 ± 2 Ma reported for the Pennsylvanian to Cisuralian
Totoltepec Pluton (e.g., see details in Kirsch et al., 2013).

4.4    CH-0403

36 unetched apatite grains from sample CH-0403 yielded a lower intercept U–Pb age of 345 ± 10
Ma with a MSWD of 0.7 and a $P(\chi^2)$ of 0.9, whereas etched grains yielded an age of 334 ± 8 Ma
with a MSWD of 1.37 and a $P(\chi^2)$ of 0.08 (Fig. 2). These cooling dates are considerably younger
if compared to the CH-0403 emplacement age of 391 ± 8 Ma (Solari et al., 2009).



4.5    OC-1008

41 unetched apatites of OC-1008 yielded a U–Pb age of 839 ± 12 Ma with a MSWD of 0.98 and
a $P(\chi^2)$ of 0.50. After etching, the same apatites yielded an age of 830 ± 10 Ma with a MSWD of
1.24 and a $P(\chi^2)$ of 0.14 (Fig. 2). Both these apatite U–Pb ages are significantly younger than the
age of granulite facies metamorphism in the Grenville-aged Oaxacan Complex (1 Ga to 980 Ma,
Solari et al., 2014), and thus, can be considered as cooling ages.
--- **Figure 2** ---


**5    Discussion and concluding remarks**

Most samples, except OV-0421, yielded slightly younger apatite U–Pb dates after etching (up to
3.3% in sample CH-0403). The lower intercept U–Pb ages obtained from unetched apatite grains
are identical within errors to the U–Pb ages obtained on the same apatites etched (see diagram in
Fig. 3). The results of our experimental study demonstrate that the chemical etching, required for
the AFT analysis, has no important effect on the determination of apatite U–Pb ages by LA–ICP-
MS. Thus, as a conclusion of this work, LA–ICP-MS can be used safely to obtain simultaneously
AFT and U–Pb ages (i.e., double dating), as it was already done in some studies without previous
proof (Chew and Donelick, 2012; Liu et al., 2014; Glorie et al., 2017; Bonilla et al., 2020; Nieto-
Samaniego et al., 2020).
--- **Figure 3** ---




**Supplement**

The supplement related to this article is available online at: https://...


**Author contributions**

Conceptualisation, investigation, and writing of the original draft were done by FA. LS and COO
provided technical support. LS and JS acquired funding and resources, supervised the study, and
reviewed the manuscript.

**Competing interests**

The authors declare that they have no conflict of interest.


**Acknowledgements**

The authors are very grateful to Juan Tomás Vázquez Ramírez and Ofelia Pérez Arvizu for their
help with sample preparation for this study. Professor Stuart Thomson (University of Arizona) is
acknowledged for sharing the Madagascar apatite. Michelangelo Martini kindly provided granite
sample TO-AM that was very useful for our experimental study.

**Financial support**

This research has been supported by PAPIIT DGAPA UNAM (grant no. IN101520 to LS).

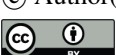





**Figure caption**


**Figure 1**
Illustration displaying the LA–ICP-MS-based U–Pb dating of the same apatite crystal before and
after chemical etching (i.e., etched in 5.5M nitric acid at 21 °C for 20 s). Spot diameter of 60 µm.


**Figure 2**
Tera–Wasserburg Concordia diagrams for the U–Pb results of unetched and etched apatites from
samples OV-0421, MCH-38, TO-AM, CH-0403, and OC-1008. MSWD – mean square weighted
deviation, Ngr – number of grains dated.


**Figure 3**
Binary plot showing the lower intercept U–Pb ages obtained on unetched and etched apatites.









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



**Figure 1**

LA-ICP-MS apatite U-Pb dating before etching

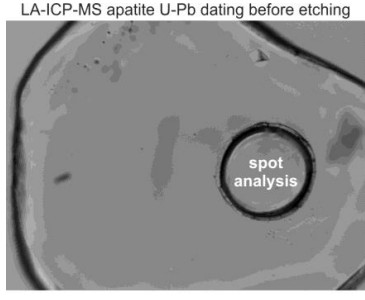

chemical etching (5.5M nitric acid, 21 °C for 20 s)

LA-ICP-MS apatite U-Pb dating after etching

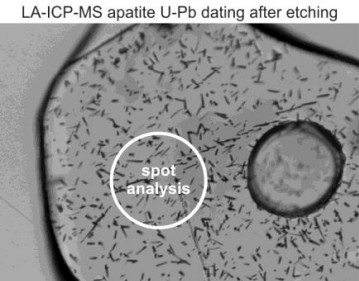














**Figure 2**

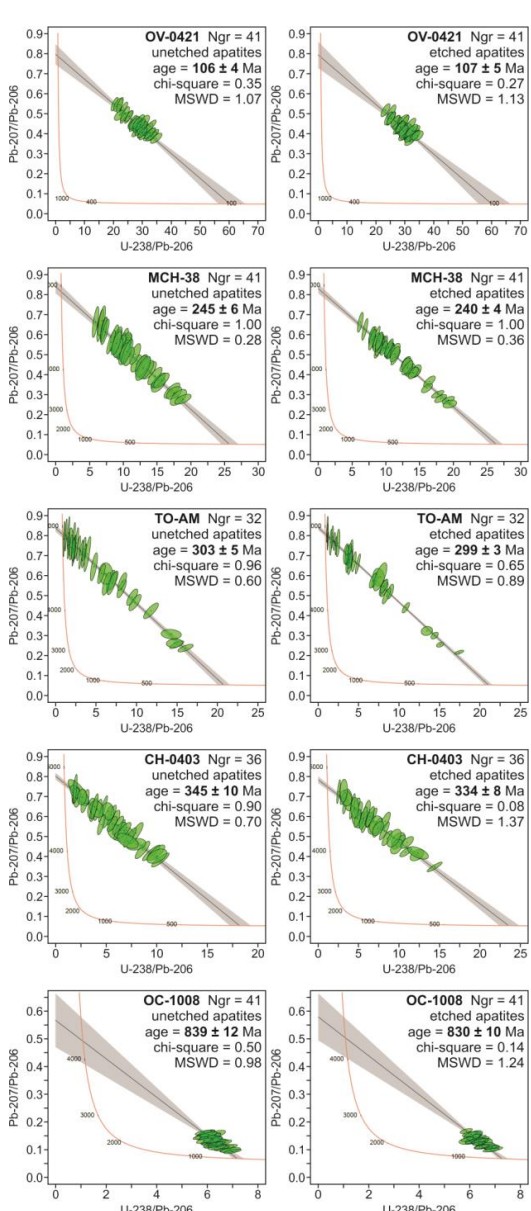









**Figure 3**

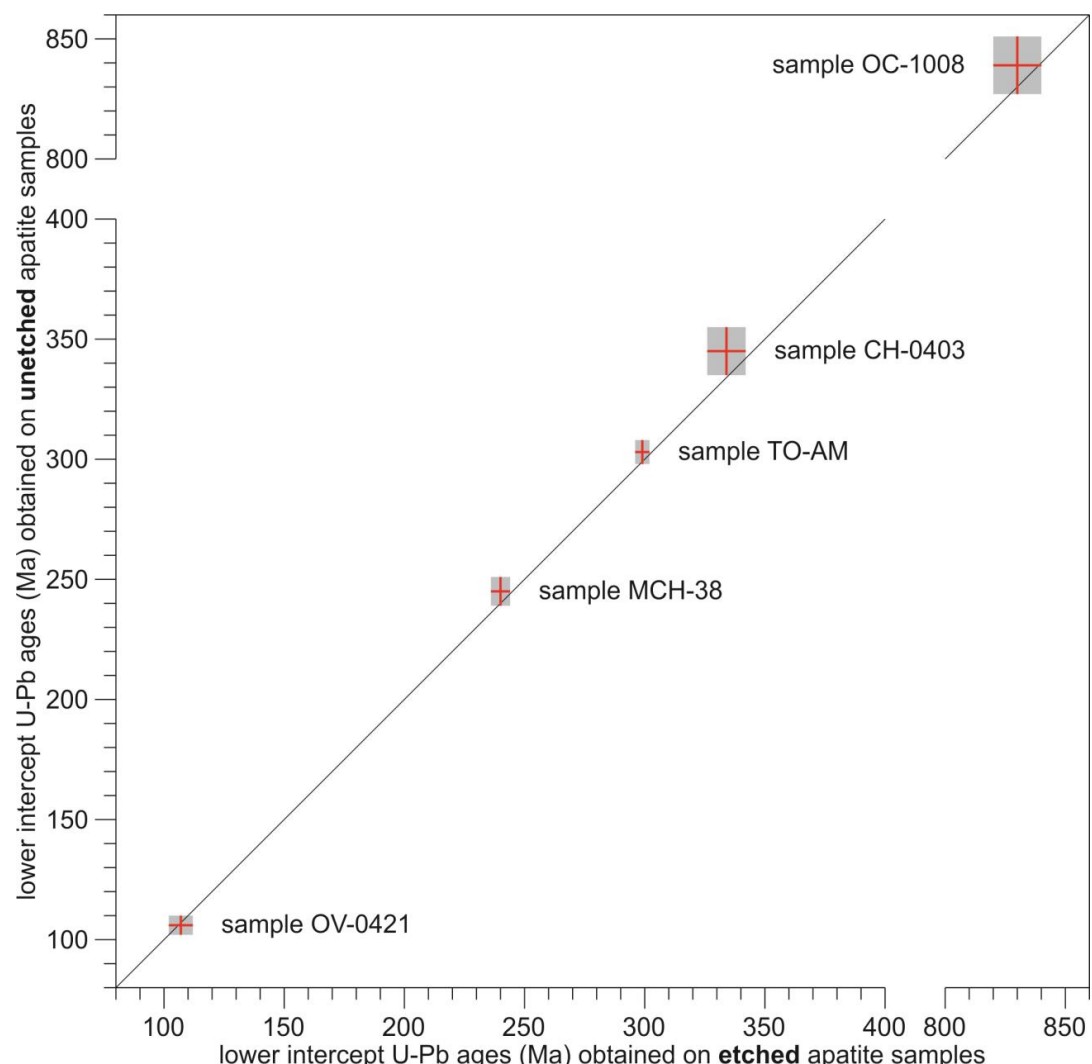












**Table 1**

Lithology, locality, and zircon U–Pb data for the selected experimental rock samples.

| Sample | Unit and locality | Rock type | Zircon U–Pb age | References |
|---|---|---|---|---|
| OV-0421 | Tres Sabanas Pluton, Guatemala | deformed granite | 115 ± 4 Ma | Torres de León (2016) |
| MCH-38 | Chiapas Massif Complex, Mexico | orthogneiss | ca. 260 to ca. 252 Ma (?) | Weber et al. (2007, 2008) |
| TO-AM | Totoltepec Pluton, Mexico | granite | ca. 308 to ca. 285 Ma (?) | Kirsch et al. (2013) |
| CH-0403 | Altos Cuchumatanes, Guatemala | granodiorite | 391 ± 8 Ma | Solari et al. (2009) |
| OC-1008 | Oaxacan Complex, Mexico | paragneiss | 990 ± 10 Ma | Solari et al. (2014) |





















**Table 2**

LA–ICP-MS protocol established at LEI to be applied for simultaneous apatite U–Pb and fission-
track *in-situ* double dating plus multielemental analysis (REEs, Y, Sr, Mn, Mg, Th, U, and Cl).

| *ICP-MS operating conditions* | |
| --- | --- |
| Instrument | Thermo Scientific™ iCAP™ Qc |
| Forward power | 1450 W |
| Carrier gas flow rate | ~1 L/min (Ar) and ~0.35 L/min (He) |
| Auxiliary gas flow rate | ~1 L/min |
| Plasma gas flow rate | ~14 L/min |
| Nitrogen | ~3.5 mL/min |
| *Data acquisition parameters* | |
| Mode of operating | STD (standard mode) |
| Sampling scheme | –2NIST-612–2MAD–1DUR–10apts– |
| Background scanning | 15 s |
| Data acquisition time | 35 s |
| Wash-out time | 15 s |
| Measured isotopes | $^{43}$Ca $^{44}$Ca $^{31}$P $^{35}$Cl $^{26}$Mg $^{55}$Mn $^{88}$Sr $^{89}$Y $^{139}$La $^{140}$Ce $^{141}$Pr $^{146}$Nd $^{147}$Sm $^{153}$Eu $^{157}$Gd $^{159}$Tb $^{163}$Dy $^{165}$Ho $^{166}$Er $^{169}$Tm $^{172}$Yb $^{175}$Lu $^{232}$Th $^{238}$U $^{204}$Pb $^{206}$Pb $^{207}$Pb $^{208}$Pb $^{202}$Hg   [total = 29] |
| *Laser ablation system* | |
| Ablation cell | RESOlution™ Laurin Technic *S-155* |
| Model of laser | Resonetics RESOlution™ *LPX Pro* |
| Wavelength | 193 nm (Excimer ArF) |
| Repetition rate | 4 Hz |
| Energy density | *4 J/cm$^2$ |
| Mode of sampling | spot diameter of 60 μm |


Note: MAD – "First mine Discovery" U–Pb apatite standard from Madagascar; DUR – Durango
apatite from Cerro de Mercado mine (Mexico); apts – unknown apatites. (*) Constant laser pulse
energy of 4 J/cm$^2$, which was measured directly on target with a Coherent™ laser energy meter.