# Peer review of "Technical note: LA–ICP-MS U–Pb dating of unetched and etched apatites"

_Geochronology, 2020_

## Referee Comment (RC1) · Jakub Sliwinski (Referee) · 7 Aug 2020

Zurich, 07.08.2020 Peer review of "Technical note: on LA–ICP-MS U–Pb dating of unetched and etched apatites" by Fanis et al.

The authors explore whether chemical etching of apatites for AFT has any influence on subsequent U-Pb dates, concluding that although etched samples tend to be a bit young compared to unetched samples, the results are well within uncertainty. The study is straightforward and the message is clear, so I have very few comments, apart from noting that the presentation needs to be cleared up in places to avoid ambiguity.

General comments:

1. Perhaps the most substantial comment: While this study demonstrates an important

effect, it does not address the fact that a very similar experiment was already undertaken by Hasebe et al., 2009 looking only at U concentrations. While I see this citation in the introduction for the very general concept of AFT, I do not see any other recognition, or any motivation explaining why this present study was undertaken. Furthermore, I see no discussion or comparison with Hasebe's study in the discussion.

2. Already in the abstract I see a few grammatical mistakes, and would therefore strongly recommend a friendly review by a colleague who is a native English speaker. Most of these mistakes are minor (misuse of articles, e.g. "the etching" instead of "etching"), but correcting them will improve the quality of the manuscript.

3. I find the abstract a little bit disappointing. While I normally enjoy concise writing, I find that a substantial part of the abstract is just "LA-ICP-MS" written out in full, and there is a lack of summary statistics for the analyses that would provide a quick and easy summary of the main results. Furthermore, given how short the abstract is, the "Short Summary" afterwards is completely redundant!

4. When reporting the ages and uncertainties (perhaps as early as the abstract), please note clearly if you're using 1s or 2s uncertainties.

5. In Hasebe, 2004 there is a short note on the potential effect of etching on LA-ICP-MS of apatites. While you show no significant difference between etched and unetched grained, the fact that you note a slight young bias makes me curious. I've worked a lot with chemical abrasion of zircons, and while the abrasion process generally removes areas of Pb loss (making the zircons older), the annealing process actually reinforces the matrix and makes the zircon look younger. This is why we always normalize abraded zircons to abraded standards. In iolite, you can actually visualize this with the time-resolved integration and see that the down-hole Pb/U fractionation is more prominent in radiation-damaged, unannealed zircons. I'd be really curious to see a down-hole fractionation signal for apatites, as this would help to determine if the slight younging is indicative of some sort of matrix-damaging process, or if it is purely

due to statistical chance. This is entirely optional, however (only for my own curiosity), so I leave it to the authors to include it or not.   Detailed comments

(format: pagenumber_line)

2_30: also U-Th dating!

2_39: I find the structure of this paragraph a bit confusing and ambiguous. Please be very clear in saying that LA-ICP-MS can be used to obtain U concentrations for AFT, as well as U-Pb ratios for U-Pb dating.

Also, I don't understand the sentence "therefore, there is a doubt..." I don't see how the doubt follows what you previously wrote.

4_87: perhaps note very quickly which "conventional" techniques you used (e.g. bromoform? Methylene iodide? Frantz?)

4_89: combine these two sentences.

4_90: what is 4pi geometry?

Table 2: Excellent table with a summary of analytical parameters. I would just note that the masses can be arranged by mass (with 238U at the end).

Figure 1: The third panel is likely unnecessary here...it's just the second panel copied and labelled with another spot location.

Figure 2: The aspect ratio of this figure is going to ruin its quality in the final print of the manuscript (i.e. it's too long to fit on a single page). Please consider splitting it into two parts, with a 3x2 grid and a 2x2 grid for two separate pages.

Also, what uncertainty is reported? 1s or 2s?

Figure 3: Just a suggestion, but maybe try plotting in log space in order to minimize the amount of blank space in the figure?

Are these error bars 1s or 2s?

Interactive
comment

---

## Referee Comment (RC2) · Ziva Shulaker (Referee) · 3 Sep 2020

This work investigates whether etching for apatite fission track dating affects the precision and uncertainty of same-grain U-Pb ages obtained via LA-ICP-MS. The authors conclude that U-Pb ages of etched and unetched apatite grains are within error of each other. However, etched grains tend to have slightly younger U-Pb ages compared to unetched grains. The purpose of the study is clear and is presented simply and understandably. However, clarification of some sentences and additional discussion would strengthen the gap in knowledge this study is filling.

Below I present the main points and minor points that require attention for revision. The major points are divided into scientific comments and the paper organization and

content. The minor comments are provided in bullet form, line-by-line. I hope the comments below are useful for ensuring that the key findings of the study are highlighted.

Major comments:

- A section summarizing previously published work, on apatite and/or zircon, and the necessity of this study should be presented before the sample description section. This will emphasize what gap in knowledge this study is filling.

- Because grains were mounted in a polished epoxy mount, it would be interesting to see if there is evidence for zoning in Cathodoluminescence (CL) or Back-scattered Electron (BSE) imagry. This could be a variable that impacts the collected U-Pb ages.

- There could be additional discussion between etched and not etched apatite U-Pb ages: to further discuss the differences between etched and not etched apatite U-Pb ages, perhaps discuss the average errors on individual U-Pb analyses for each sample. Often in U-Pb geochronology, individual U-Pb analyses can have high errors but the reported weighted mean age and errors can result in an age with a severely underestimated error. This could therefore mask whether U-Pb analyses on etched grains are more imprecise or less accurate compared to unetched grains.

- Are there noticeable differences between Th, Pb, and/or U concentrations collected via LA-ICP-MS before or after etching apatite grains? Or do these grains have very variable Th, Pb, and/or U concentrations? Does elemental concentration affect ages determined after etching? Homogeneous standards could help assess these points.

- An increasing number of studies couple same-grain multi-analytical techniques to obtain as much information as possible. For instance, performing (U-Th)/He and/or U-Pb and/or trace-element analyses on zircons or apatites. It would be interesting to discuss the effects of apatite fission track etching with U-Pb and trace-elements. I am unsure whether additional trace-elements were collected in this study, as the protocol that was used in this study is stated to have been developed for U-Pb and multi-element analy-
ses (line 101). If this data exists, I think this discussion could enhance the applicability and reach of this manuscript.

Minor comments:

- Switch the first and second sentences so that the objective occurs first and then the experiment is discussed.

- Lines 11-13: incorrect grammar; also clarify the "obtaining" of U-Pb ages; perhaps replace with something similar to as follows: "The objective of this study is to assert whether etching required for apatite fission track analyses impacts the precision and accuracy of same-grain U-Pb ages."

- Line 14: determination of apatite U-Pb ages is vague; clarify "determination" – such as accuracy, precision?

- Line 16: instead of simultaneously; "double dating" is more accurate. I interpret the goal of this sentence to assert that this paper establishes the viability of double dating apatite via fission track and LA-ICP-MS.

- Line 19: "of five samples" should be replaced with "from five samples"

- Line 21: clarify – obtaining accurate and precise U-Pb ages?

- Lines 29-30: "This accessory mineral is often used for fission track, (U-Th)/He, and U-Pb dating"

- Lines 33-34: acronym should come after the term: eg., "laser ablation inductively coupled plasma mass spectrometry (LA-ICP-MS)"

- Line 35: clarify 238U levels - concentration?

- Line 40: clarify what causes the doubt and what the doubt is.

- Lines 40-43: (my personal preference is to avoid asking questions) could restate questions as: "The influence of chemical etching required for AFT dating of the precision and accuracy of same-grains analyzed for U-Pb dating via LA-ICP-MS remains to be quantified. To investigate this issue, the same unetched and etched apatite grains were analyzed via LA-ICP-MS for U-Pb dating."

- Line 47: header can be simpler, such as: Sample descriptions

- Lines 59-62: validate using the age of a different previously dated sample; where is the other dated sample in relation to the sample in this study? Is it from the same unit?

- Lines 87-90: combining sentences for reading ease: "Approximately 300 apatite grains were extracted from each rock sample and mounted with their surfaces parallel to the crystallographic c-axis in a 2.5 cm diameter epoxy mount. The mount was polished. . ."

- Line 91: "sterile" is unclear; sufficient to state: "For our experiment, complete crystals lacking visible inclusions and other defects, such as cracks, were selected for analysis."

- Line 97: remove "exactly" unless the center of the polished surface was measured for spot analysis

- Line 101: were other elements (REEs, Y, Sr, Mn, Mg, Cl) measured in this study? Or the same protocol that was developed to measure those elements was used? If they weren't measured in this study, I would disregard from Table 2.

- Line 103: include the Iolite version

- Line 122: should be moved to the analytical procedures in the paragraph beginning line 94

- Lines 176-177: this contradicts the first sentence of the paragraph; there is clearly some effect to the U-Pb ages after etching, but it might be within analytical uncertainty and grains analyzed before/after etching have indistinguishable U-Pb ages.

- Line 178: word choices of "safely" and "simultaneous;" perhaps restate sentence to describe how this work shows that chemical etching for AFT dating doesn't significantly

affect U and Pb ratios or concentrations, which makes apatite grains analyzed for AFT amenable to same-grain U-Pb dating via LA-ICP-MS.

- Lines 179-181: this should be in a new section above that discusses previous work. This will emphasize why your study is vital for providing data that validates same-grain AFT and apatite U-Pb dating via LA-ICP-MS.

- Line 217: in the Figure 2 caption, note whether the ages reported are averages, weighted means, etc.; are the uncertainties one or two sigma?

- Line 223: in figure 3, what are the errors shown on the graph?

- Line 455: see comment for line 101: if additional elements (REEs, Y, Sr, Mn, Mg, Cl) were not measured, can disregard from table 2 as this wasn't the set-up for these experiments.

---

## Author Comment (AC1) · 16 Oct 2020

**Response to Referee 1 – Jakub Sliwinski**

The authors explore whether chemical etching of apatites for AFT has any influence on subsequent U-Pb dates, concluding that although etched samples tend to be a bit young compared to unetched samples, the results are well within uncertainty. The study is straightforward, and the message is clear, so I have very few comments, apart from noting that the presentation needs to be cleared up in places to avoid ambiguity.

*Dear Jakub, thank you for your comments and reviewing our manuscript. Please, find our responses (in red italics) to each of your comments below.*

General comments:

1. Perhaps the most substantial comment: While this study demonstrates an important effect, it does not address the fact that a very similar experiment was already undertaken by Hasebe et al., 2009 looking only at U concentrations. While I see this citation in the introduction for the very general concept of AFT, I do not see any other recognition, or any motivation explaining why this present study was undertaken. Furthermore, I see no discussion or comparison with Hasebe's study in the discussion.

*OK, done. We compared and mentioned Hasebe's study.*

2. Already in the abstract I see a few grammatical mistakes and would therefore strongly recommend a friendly review by a colleague who is a native English speaker. Most of these mistakes are minor (misuse of articles, e.g. "the etching" instead of "etching") but correcting them will improve the quality of the manuscript.

*OK, done. Thank You. In our view, the grammar was improved.*

3. I find the abstract a little bit disappointing. While I normally enjoy concise writing, I find that a substantial part of the abstract is just "LA-ICP-MS" written out in full, and there is a lack of summary statistics for the analyses that would provide a quick and easy summary of the main results. Furthermore, given how short the abstract is, the "Short Summary" afterwards is completely redundant!

*OK. Abstract was improved. Short Summary is required by the Journal.?*

4. When reporting the ages and uncertainties (perhaps as early as the abstract), please note clearly if you're using 1s or 2s uncertainties.

*OK, done.*

5. In Hasebe, 2004 there is a short note on the potential effect of etching on LA-ICP-MS of apatites. While you show no significant difference between etched and unetched grained, the fact that you note a slight young bias makes me curious. I've worked a lot with chemical abrasion of zircons, and while the abrasion process generally removes areas of Pb loss (making the zircons older), the annealing process actually reinforces the matrix and makes the zircon look younger. This is why we always normalize abraded zircons to abraded standards. In iolite, you can actually visualize this with the time-resolved integration and see that the down-hole Pb/U fractionation is more prominent in radiation-damaged, unannealed zircons. I'd be really curious to see a down-hole fractionation signal for apatites, as this would help to determine if the slight youngling is indicative of some sort of matrix-damaging process, or if it is purely due to statistical chance. This is entirely optional, however (only for my own curiosity), so I leave it to the authors to include it or not.

*Thank You for recommendation. You are right, U-Pb ages are slightly younger on etched apatite crystals. This was now discussed.*

Detailed comments

(format: page number_line)

2_30: also U-Th dating!

*OK, Thank You.*

2_39: I find the structure of this paragraph a bit confusing and ambiguous. Please be very clear in saying that LA-ICP-MS can be used to obtain U concentrations for AFT, as well as U-Pb ratios for U-Pb dating. Also, I don't understand the sentence "therefore, there is a doubt . . ." I don't see how the doubt follows what you previously wrote.

*OK, done. The sentence was improved.*

4_87: perhaps note very quickly which "conventional" techniques you used (e.g. bromoform? Methylene iodide? Frantz?)

*OK, done. We used sieving, Frantz, and bromoform.*

4_89: combine these two sentences.

*OK, done. Thank You.*

4_90: what is 4pi geometry?

*4pi geometry is generally used for AFT dating. 4pi geometry referees to polishing up to the interior of crystals (e.g., for apatites, removing 15-20 microns, o more).*

Table 2: Excellent table with a summary of analytical parameters. I would just note that the masses can be arranged by mass (with 238U at the end).

*OK, done.*

Figure 1: The third panel is likely unnecessary here . . . it's just the second panel copied and labelled with another spot location.

*OK. The third panel was removed.*

Figure 2: The aspect ratio of this figure is going to ruin its quality in the final print of the manuscript (i.e. it's too long to fit on a single page). Please consider splitting it into two parts, with a 3x2 grid and a 2x2 grid for two separate pages. Also, what uncertainty is reported? 1s or 2s?

*Ok, done.*

Figure 3: Just a suggestion, but maybe try plotting in log space in order to minimize the amount of blank space in the figure? Are these error bars 1s or 2s?

*I think if we plot in log scale, we cannot see well error bars. 2SE bars.*

---

## Author Comment (AC2) · 16 Oct 2020

**Response to Referee 2 – Ziva Shulaker**

This work investigates whether etching for apatite fission track dating affects the precision and uncertainty of same-grain U-Pb ages obtained via LA-ICP-MS. The authors conclude that U-Pb ages of etched and unetched apatite grains are within error of each other. However, etched grains tend to have slightly younger U-Pb ages compared to unetched grains. The purpose of the study is clear and is presented simply and understandably. However, clarification of some sentences and additional discussion would strengthen the gap in knowledge this study is filling.

Below I present the main points and minor points that require attention for revision. The major points are divided into scientific comments and the paper organization and content. The minor comments are provided in bullet form, line-by-line. I hope the comments below are useful for ensuring that the key findings of the study are highlighted.

*Dear Ziva, thank you very much for your comments and reviewing this paper. Please, find our responses (in red italics) to each of your comments below.*

Major comments:

- A section summarizing previously published work, on apatite and/or zircon, and the necessity of this study should be presented before the sample description section. This will emphasize what gap in knowledge this study is filling.

*OK, done. The section of Introduction was improved, and the importance of our study is now presented in a clearer form.*

- Because grains were mounted in a polished epoxy mount, it would be interesting to see if there is evidence for zoning in Cathodoluminescence (CL) or Back-scattered Electron (BSE) imagery. This could be a variable that impacts the collected U-Pb ages.

*Unfortunately, we are unable to obtain CL neither BSE images. On the other hand, apatite grains analyzed in this study show no significant variation on elemental composition. We compared REEs and trace elements on same unetched and etched apatite grains and noted that there are no significant differences.*

- There could be additional discussion between etched and not etched apatite U-Pb ages: to further discuss the differences between etched and not etched apatite U-Pb ages, perhaps discuss the average errors on individual U-Pb analyses for each sample. Often in U-Pb geochronology, individual U-Pb analyses can have high errors but the reported weighted mean age and errors can result in an age with a severely underestimated error. This could therefore mask whether U-Pb analyses on etched grains are more imprecise or less accurate compared to unetched grains.

*In our view, it is not necessary to compare single-grain U-Pb ages between unetched and etched grains, because apatite U-Pb ages obtained by LA-ICP-MS are generally discordant. You are right, etched grains apparently yielded more precise ages if compared to unetched grains. This now was also discussed.*

- Are there noticeable differences between Th, Pb, and/or U concentrations collected via LA-ICP-MS before or after etching apatite grains? Or do these grains have very variable Th, Pb, and/or U concentrations? Does elemental concentration affect ages determined after etching? Homogeneous standards could help assess these points.

- An increasing number of studies couple same-grain multi-analytical techniques to obtain as much information as possible. For instance, performing (U-Th)/He and/or U-Pb and/or trace-element analyses on zircons or apatites. It would be interesting to discuss the effects of apatite fission track etching with U-Pb and trace-elements. I am unsure whether additional trace-elements were collected in this study, as the protocol that was used in this study is stated to have been developed for U-Pb and multi-element analyses (line 101). If this data exists, I think this discussion could enhance the applicability and reach of this manuscript.

*We revised carefully Th, U, REE, Sr, Y, Mn, Mg, an Cl contents before and after etching the same crystals. There are no marked differences. Pooled concentrations are identical between unetched and etched apatite groups from each sample.*

Minor comments:

- Switch the first and second sentences so that the objective occurs first and then the experiment is discussed.

*OK. Done.*

- Lines 11-13: incorrect grammar; also clarify the "obtaining" of U-Pb ages; perhaps replace with something similar to as follows: "The objective of this study is to assert whether etching required for apatite fission track analyses impacts the precision and accuracy of same-grain U-Pb ages."

*OK. Done. Thank You.*

- Line 14: determination of apatite U-Pb ages is vague; clarify "determination" – such as accuracy, precision?

*OK. Done. Thank You.*

- Line 16: instead of simultaneously; "double dating" is more accurate. I interpret the goal of this sentence to assert that this paper establishes the viability of double dating apatite via fission track and LA-ICP-MS.

*OK. Done.*

- Line 19: "of five samples" should be replaced with "from five samples"

*OK. Done. Thank You.*

- Line 21: clarify – obtaining accurate and precise U-Pb ages?

*OK. Done.*

- Lines 29-30: "This accessory mineral is often used for fission track, (U-Th)/He, and U-Pb dating"

*OK. Done.*

- Lines 33-34: acronym should come after the term: eg., "laser ablation inductively coupled plasma mass spectrometry (LA-ICP-MS)"

*OK. Done. Thank You.*

- Line 35: clarify 238U levels - concentration?

*OK. Done. "concentrations".*

- Line 40: clarify what causes the doubt and what the doubt is.

*OK. Done. Thank You.*

- Lines 40-43: (my personal preference is to avoid asking questions) could restate questions as: "The influence of chemical etching required for AFT dating of the precision and accuracy of same-grains analyzed for U-Pb dating via LA-ICP-MS remains to be quantified. To investigate this issue, the same unetched and etched apatite grains were analyzed via LA-ICP-MS for U-Pb dating."

*OK. Thank You. The sentence was improved as You suggested.*

- Line 47: header can be simpler, such as: Sample descriptions

*OK.*

- Lines 59-62: validate using the age of a different previously dated sample; where is the other dated sample in relation to the sample in this study? Is it from the same unit?

*OK. Sample MCH-38 is from the same unit.*

- Lines 87-90: combining sentences for reading ease: "Approximately 300 apatite grains were extracted from each rock sample and mounted with their surfaces parallel to the crystallographic c-axis in a 2.5 cm diameter epoxy mount. The mount was polished..."

*OK. Done. Thank You.*

- Line 91: "sterile" is unclear; sufficient to state: "For our experiment, complete crystals lacking visible inclusions and other defects, such as cracks, were selected for analysis."

*OK. Done. Thank You.*

- Line 97: remove "exactly" unless the center of the polished surface was measured for spot analysis

*OK.*

- Line 101: were other elements (REEs, Y, Sr, Mn, Mg, Cl) measured in this study? Or the same protocol that was developed to measure those elements was used? If they weren't measured in this study, I would disregard from Table 2.

*Yes, all these mases were measured during this study. I think it is important to demonstrate this protocol, which may be useful for further experiments.*

- Line 103: include the Iolite version

*OK. Done.*

- Line 122: should be moved to the analytical procedures in the paragraph beginning line 94

*OK. Done.*

- Lines 176-177: this contradicts the first sentence of the paragraph; there is clearly some effect to the U-Pb ages after etching, but it might be within analytical uncertainty and grains analyzed before/after etching have indistinguishable U-Pb ages.

*OK. Done. The paragraph was improved and clarified.*

- Line 178: word choices of "safely" and "simultaneous;" perhaps restate sentence to describe how this work shows that chemical etching for AFT dating doesn't significantly affect U and Pb ratios or concentrations, which makes apatite grains analyzed for AFT amenable to same-grain U-Pb dating via LA-ICP-MS.

*OK. Done.*

- Lines 179-181: this should be in a new section above that discusses previous work. This will emphasize why your study is vital for providing data that validates same-grain AFT and apatite U-Pb dating via LA-ICP-MS.

*OK. Done. Thank You.*

- Line 217: in the Figure 2 caption, note whether the ages reported are averages, weighted means, etc.; are the uncertainties one or two sigma?

*OK. Done.*

- Line 223: in figure 3, what are the errors shown on the graph?

*OK. Done.*

- Line 455: see comment for line 101: if additional elements (REEs, Y, Sr, Mn, Mg, Cl) were not measured, can disregard from table 2 as this wasn't the set-up for these experiments.

*We think it is important to demonstrate this protocol in this manuscript.*

---

## Author Response (AR2)

[revised manuscript text omitted]

---

## Author Response (AR3)

[revised manuscript text omitted]

*Letter for Professor Klaus Mezger*

                                                                              *Santiago de Querétaro, 15 Nov 2020*

**Dear Professor Klaus Mezger,**

We revised our manuscript according to your minor comments.

Below, I attached a pdf.file with track changes.

**With Best Wishes,**

Fanis

[revised manuscript text omitted]